Evaluating manta ray mucus as an alternative DNA source for population genetics study: underwater-sampling, dry-storage and PCR success

Kashiwagi Tom 1 2 4 tomkashi@gmail.com
Maxwell Elisabeth A. 3
Marshall Andrea D. 2
Christensen Ana B. 3
1 Molecular Fisheries Laboratory, University of Queensland , St. Lucia, QLD , Australia
2 Marine Megafauna Foundation , Truckee, CA , USA
3 Biology Department, Lamar University , Beaumont, TX , USA
4 Current affiliation: Center for Fisheries, Aquaculture and Aquatic Sciences, Southern Illinois University Carbondale , Carbondale, IL , USA
Esteban María Ángeles
Electronic publication date: 2015 Aug 13
Publication date: 2015
Volume: 3
Electronic Location ID: e1188
Received 2014 Dec 15; Accepted 2015 Jul 23
Copyright: © 2015 Kashiwagi et al.
Copyright year: 2015
Copyright holder: Kashiwagi et al.
License: This is an open access article distributed under the terms of the Creative Commons Attribution License, which permits unrestricted use, distribution, reproduction and adaptation in any medium and for any purpose provided that it is properly attributed. For attribution, the original author(s), title, publication source (PeerJ) and either DOI or URL of the article must be cited.
License URL: https://creativecommons.org/licenses/by/4.0/

Keywords: Animal welfare, CITES, Fish pain, Eco-tourism, Epidermal cells, CMS, Whole genome amplification, Stable isotope, Genotyping errors, SCUBA

Funding: David J. Beck Fellowship Elisabeth A. Maxwell was supported by the David J. Beck Fellowship. The funders had no role in study design, data collection and analysis, decision to publish, or preparation of the manuscript.

==============================
Sharks and rays are increasingly being identified as high-risk species for extinction, prompting urgent assessments of their local or regional populations. Advanced genetic analyses can contribute relevant information on effective population size and connectivity among populations although acquiring sufficient regional sample sizes can be challenging. DNA is typically amplified from tissue samples which are collected by hand spears with modified biopsy punch tips. This technique is not always popular due mainly to a perception that invasive sampling might harm the rays, change their behaviour, or have a negative impact on tourism. To explore alternative methods, we evaluated the yields and PCR success of DNA template prepared from the manta ray mucus collected underwater and captured and stored on a Whatman FTA™ Elute card. The pilot study demonstrated that mucus can be effectively collected underwater using toothbrush. DNA stored on cards was found to be reliable for PCR-based population genetics studies. We successfully amplified mtDNA ND5, nuclear DNA RAG1, and microsatellite loci for all samples and confirmed sequences and genotypes being those of target species. As the yields of DNA with the tested method were low, further improvements are desirable for assays that may require larger amounts of DNA, such as population genomic studies using emerging next-gen sequencing.

Introduction

Sharks and rays are increasingly being identified as high-risk species for extinction, prompting urgent assessments of their local or regional populations (Dulvy et al., 2014a). The Reef Manta Ray Manta alfredi (Krefft 1868) and the Giant Manta Ray M. birostris (Walbaum 1792) are currently listed as Vulnerable by the International Union for the Conservation of Nature (IUCN) Red List of Threatened Species in 2011 (Marshall et al., 2011a; Marshall et al., 2011b). Both species have been listed on Appendix I & II of the Convention for Migratory Species (CMS) and both species were recently awarded Appendix II listing on the Conventions on International Trade in Endangered Species of Wild Fauna and Flora (CITES). These key conservation steps represent the first significant movement to address reported global declines in manta rays (Vincent et al., 2014). Manta rays have been described as having extremely conservative life history traits, representing one of the least fecund elasmobranch species and with one of the lowest maximum intrinsic rates of population increase of any studied Chondrichthyan (Couturier et al., 2012; Dulvy et al., 2014b).

A crucial knowledge gap still exists in the empirical understanding of their population dynamics, structure, status and trends, which needs to be addressed for the implementation of effective management (CITES, 2013). DNA-based population studies can complement logistically and financially challenging long-term field studies by providing insights into the patterns of population structure, connectivity, and effective population sizes (Dudgeon et al., 2012; Schwartz, Luikart & Waples, 2007).

Apart from land-based sampling at fish landing sites, manta ray tissue samples are typically collected underwater while SCUBA or free diving using hand spears with biopsy punch tips (Fig. 1). As manta rays are a major attraction for tourism (O’Malley, Lee-Brooks & Medd, 2013), such sampling activity may not be popular or discouraged in some areas where people fear that the technique might harm the rays, change their behaviour or have a negative impact on tourism (Braithwaite, 2010; Huntingford et al., 2006; Rose et al., 2014). Availability of alternative and less invasive methods to collect DNA from manta rays would increase sampling opportunities.

Figure 1 Tissue sampling with a biopsy tip and a hand spear.

Here we test the feasibility of the collection of body surface mucus from wild manta rays and its effectiveness as a DNA source for PCR-based population genetics studies. Epidermal cells in surface mucus have been successfully used in many studies for humans, livestock, and wild animals (Gustavsson et al., 2009; Le Vin et al., 2011; McClure et al., 2009; Prunier et al., 2012; Smith & Burgoyne, 2004), but only a handful of studies exist that have examined large marine fish (Hoolihan et al., 2009; Lieber et al., 2013). Lieber et al. (2013), recently reported an ∼75% PCR success rate using mucus from the Basking Shark Cetorhinus maximus (Gunnerus 1765), stored in 99 % ethanol, in amplifying the high copy number mitochondrial DNA (mtDNA) genes cytochrome c oxidase subunit 1 (CO1) and control region (CR) and the nuclear ribosomal internal transcribed spacer 2 (ITS2) region. The feasibility of using mucus from other sharks and rays has been largely unexplored, particularly in regards to underwater collection, amplification of single copy nuclear genes and microsatellites, and dry storage methods that may eliminate the needs for special shipping considerations and freezers (Smith & Burgoyne, 2004; Williams, 2007). Here we report preliminary results on the effectiveness of these techniques, limitations, and its applicability to future manta ray research. We also discuss potential areas for improvement and future directions.

Materials and Methods

All procedures were conducted in accordance to the University of Queensland Animal Ethics Committee approval number SBMS/206/11/ARC and Ecuadorian Ministry of the Environment research permits: 009RM-DPM-MA.

Mucus from eighteen Manta birostris was collected on SCUBA from Isla de la Plata in Ecuador (1°15 29.62S, 81°4 25.96W) between 2 September and 20 September 2012. Samples were obtained using a small toothbrush held in the diver’s hand (Video S1) or mounted on an extendable pole (Fig. 2). For each sample, the dorsal surface of the ray was rubbed back and forth or in a circular motion ∼3–5 times, then the brush was placed into an individual 50 ml plastic tube to prevent cross contamination. On dry land, approximately 120 µl mucus was transferred from the brush with a clean sterile cotton bud and then onto FTA™ Elute Cards and/or Indicating FTA™ Elute Cards (GE Healthcare) using three side-to-side motions, 90° each way (Fig. 3), spreading mucus and cells evenly to an area of approximately 625 mm2. These cards, which are impregnated with a chemical formula that lyses cells and denatures proteins upon contact, are designed for room temperature storage and shipment of DNA from biological samples for PCR analysis. The applied volume of liquid samples is the recommended amount to avoid overloading the chemicals (GE Healthcare). Cards were then air dried and placed in separate resealable plastic bags. Samples were then transported via land and air as normal domestic and international postage and kept at room temperature with desiccants until further analysis in the lab.

Figure 2 Mucus sampling with a toothbrush mounted on an extendable pole.

Figure 3 Application of mucus to FTA card.

(A) Black mucus collected on toothbrush, (B) cotton bud with ∼120 µl of mucus, (C) transferring mucus onto FTA card using three side-to-side motions, 90° each way, (D) FTA card with mucus sample.

DNA for downstream analyses was prepared using the recommended simple protocol for FTA™ Elute Cards that releases single stranded DNA (ssDNA) into water. Three squares (6 mm × 6 mm × 3) were cut out using a clean scalpel, washed by pulse-vortex in 1.5 ml of sterile water for 5 s, then placed in 300 µl of sterile water and heated at 98 °C for 30 min. At the end of the incubation step, tubes went through 60 times pulse-vortex at a rate approximately one pulse/second. The cut-outs were removed from tubes and eluates were stored at −20 °C until further analyses.

The quality and quantity of template DNA was assessed with three commonly used methods: Spectrophotometry (NanoDrop™ 1000; Thermo Scientific), fluorometry (Qubit™ ssDNA Assay Kit; Invitrogen), and 1% agarose gel electrophoresis. Spectrophotometry can be used to estimate DNA concentration based on light absorbance measurements at wavelength 260 nm (A260). As a number of substances also absorb light at 260 nm, the results may be unreliable or inaccurate when samples are not purified DNA. Lower detection limit of the instrument was 2 ng/µl. Fluorometry (Qubit™ ssDNA Assay Kit, Invitrogen) uses fluorescent-based dyes that bind specifically to DNA, thus more reliable for the quantification of the target molecules. Lower detection limit of the assay employed was 0.05 ng/µl. Gel electrophoresis can provide information on DNA quantity and quality. High amount and intact genomic DNA should appear as a bright compact, high-molecular weight band whereas low amount and degraded DNA might appear faint and low-molecular-weight smears.

We performed PCR for mtDNA (ND5), nuclear DNA (RAG1) and three microsatellite loci (MA09, MA14 and MA34) using published protocols and 1–4 µl of template DNA in 12–20 µl reaction (Kashiwagi et al., 2012a; Kashiwagi et al., 2012b) with positive and negative controls and replicated experiments. PCR products for ND5 and RAG1 genes were sequenced in both forward and reverse directions and compared with known types (Kashiwagi et al., 2012b) with GenBank Accession numbers FJ235624–FJ235631 and KR703213–KR703233. PCR products for three microsatellites were genotyped and compared with previously reported size range (Kashiwagi et al., 2012a). Robustness of genotyping results were tested with replicated experiments for their consistency.

Results

Time between sampling and lab analyses ranged from 81 to 343 days. DNA concentration estimated by spectrophotometric measurements of the concentration of DNA templates ranged from 12.18 to 29.00 ng/µl (23.16 ± 4.05 ng/µl, mean ± s.d., n = 18). Estimate for blank sample (i.e., card only) was 11.7 ng/µl. Absorbance spectra lacked the typical peak at wavelength 260 nm preceded by a dip at 230 nm, which was observable in DNA templates prepared from tissue samples using a commercial DNA extraction kit (e.g., Qiagen DNeasy Kit) (Fig. 4). Instead, spectra showed high absorbance around wavelength 230–240 nm that was also present in blank sample. Fluorometric measurements ranged from 0.0743 to 2.16 ng/µl (0.589 ± 0.536 ng/µl, mean ± s.d., n = 18). Measurement for blank sample (i.e., card only) was lower than the detection limit of 0.05 ng/µl. There was no visible band or smear with gel electrophoresis loaded with 10 µl of samples. Samples concentrated approximately ten times by both standard ethanol precipitation and vacuum drying also failed to show a band or smear.

Figure 4 Absorbance spectrum of DNA prepared from mucus samples (grey lines), blank (i.e., card only, black dotted) and a tissue sample with DNA extraction kit (black solid) measured by Nanodrop™.

PCR was successful for all five markers (ND5, RAG1 and three microsatellite loci) across all 18 samples. Fourteen of 18 samples showed known M. birostris ND5 haplotypes (Table 1). One sample showed haplotype (MA04) previously only known from M. alfredi. Three new haplotypes were detected (MB13, 14 and 15, GenBank Accession numbers KR703235, KR703236, and KR707237, respectively), that were within 3 bp differences in comparison to common haplotype MB01. Seventeen of 18 samples showed known M. birostris RAG1 sequence types. One sample showed new sequence type, MBRAG05 (GenBanK Accession Number KR73234), that is uniquely heterozygous R = A + G at position #615 in comparison to all other known types that are homozygous G at the position. Size range for microsatellite was 378–394 (locus MA09), 189–221 (locus MA14), and 183–189 (locus MA34), that extended previously reported ranges for these loci, 378, 197–221, and 185–189, respectively (Kashiwagi et al., 2012a). Genotyping results were consistent among replicated PCR experiments.

Table 1 Sequencing and genotyping results.

Sample name	RAG1	ND5	Microsatellite loci	
			MA09	MA09	MA14	MA14	MA34	MA34	
EC01	MBRAG01	MB06	378	378	201	209	187	187	
EC02	MBRAG01	MB05	378	378	197	201	189	189	
EC03	MBRAG01	MB06	378	378	201	209	187	187	
EC04	MBRAG05a	MB06	378	378	197	201	189	189	
EC05	MBRAG01	MB01	378	390	201	217	189	189	
EC06	MBRAG01	MB01	378	386	209	221	185	187	
EC07	MBRAG02	MA04	378	378	197	201	185	187	
EC08	MBRAG01	MB01	378	386	209	221	185	187	
EC09	MBRAG01	MB13a	378	390	197	209	187	191	
EC10	MBRAG01	MB01	378	394	193	221	187	187	
EC11	MBRAG01	MB01	378	390	189	201	187	187	
EC12	MBRAG01	MB01	386	390	201	201	187	187	
EC13	MBRAG01	MB01	378	390	197	201	187	189	
EC14	MBRAG02	MB01	378	378	197	201	187	187	
EC15	MBRAG01	MB14a	378	390	189	201	187	187	
EC16	MBRAG02	MB15a	378	378	201	221	187	187	
EC17	MBRAG02	MB01	378	378	197	201	183	187	
EC18	MBRAG02	MB05	378	378	189	201	183	187	
Notes.

a Newly detected sequence types in this study (GenBank: KR73234–KR73237). Sequence types without superscript are as Kashiwagi et al. (2012b) (GenBank: FJ235624–FJ235631 and KR703213–KR703233).

Discussion

Our results demonstrate that DNA from manta ray mucus collected underwater and stored dry on FTA™ Elute cards can be reliably used in PCR-based population genetic studies. To our knowledge, this study is the first example involving underwater collection of mucus by SCUBA divers. The advantages of the developed method include: (i) a reduction in sampling gear, (ii) a significantly reduced impact on the sampled organism, (iii) an increased acceptance as a sampling protocol in region of vibrant tourism, and (iv) reliable, dry, room temperature storage of DNA without need for liquid reagents, refrigerator/freezer, and special shipping considerations.

Sampling of manta ray mucus can be relatively easily achieved, but should only be attempted by experienced field researchers that understand the behavior of these animals. Minimal to no reaction to sampling was noted in all samples taken from manta rays in Ecuador by experienced field researchers (see Video S1). Several collection tools were initially tested to trap mucus from the dorsal and ventral surfaces of mantas including scouring pads, cotton buds, cotton wool, and a small comb, but small disposable toothbrushes were found to be most effective. Adequate amount of samples were obtained using the toothbrush in the hand or attached to an extendable pole, however the former technique was more effective in applying forces to the brush. Researchers wanting to sample individual manta rays that cannot be approached closely underwater or that are sampling manta rays at the surface from a boat may benefit from the latter technique. Larger amount of mucus samples were taken with circular brushing motions than back and forth motions and from the dorsal surface rather than the ventral surface. The black pigment on the dorsal surface (Coles, 1916) tinting the mucus helps visual confirmation of mucus on the brush during underwater collection and transferring onto the FTA Elute Cards on land.

DNA yields from FTA Elute cards using the simple purification method were low judging from the lack of visible DNA in gel electrophoresis and fluorometric measurements showing that only three out of 18 templates were above 1 ng/µl. The estimated DNA concentration by spectrophotometry is likely to be inaccurate judging from the lack of distinct peak at 260 nm in spectrometry and positive value in blank sample. This is an expected result judging from the manufacturer’s product information (GE Healthcare Life Sciences, 2012) and empirical findings (De Vargas Wolfgramm et al., 2009), which state that the single stranded DNA eluted from FTA Elute cards using the simple protocol is often below the lower detection limit of the current spectrophotometer. As such, we recommend flurometric quantitation as an important first step in downstream analyses for avoiding genotyping errors by using too little copy number of template DNA (Taberlet et al., 1996; Taberlet, Waits & Luikart, 1999). Furthermore, it is safest to assume that the amount of cell materials in a given volume of mucus is low. Therefore, it is important that the sampler spread the mucus evenly and fully onto the card. Indeed, we observed variable PCR success when preparing DNA template from three 3 mm diameter punches, where the samples were simply tapped onto the card directly from the brush (E Maxwell and A Christensen, 2013, unpublished data). We are uncertain about substances that produced high spectrophotometry reading ∼230–240 nm among mucus and blank samples from FTA card. It appears those impurities and low DNA yield did not interfere with downstream analyses in this study.

Sequencing results and genotyping results confirmed that the targeted DNA of the species was amplified. Discovery of new haplotypes, sequence types and extended size range for microsatellite is as expected because samples from Ecuador were not included in previous studies (Kashiwagi et al., 2012a; Kashiwagi et al., 2012b). Discovery of one M. birostris individual with mtDNA haplotype MA04, that is only 1 bp different from common M. birostris haplotype MB01, but previously known from M. alfredi only, was surprising but interpretable. As the speciation event of M. alfredi and M. birostris was recent, lineage sorting of various genes might not be complete and post divergence hybridization might have occured occasionally (Kashiwagi et al., 2012b). Manta alfredi do not occur in East Pacific Ocean (Kashiwagi et al., 2011; Marshall et al., 2011b) and the inspection of the photograph of the individual clearly keys out as Manta birostris (Marshall, Compagno & Bennett, 2009). Thus, there is no evidence to support the hypothesis that the observed results are due to hybridization in current generations. Our interpretation is that the results indicate that the current mtDNA marker is not yet information-rich enough to distinguish the two species world-wide.

We successfully sequenced M. alfredi using this method as well (E Maxwell and A Christensen, 2013, unpublished data). We recommend that the potential utilization of mucus samples beyond the basic PCR based assay be explored further because high quality and quantity of DNA will likely become increasingly important for population genomic analyses with emerging technological advancement in high throughput sequencing (Allendorf, Hohenlohe & Luikart, 2010; Hohenlohe, Catchen & Cresko, 2012; Narum et al., 2013). Higher yields and purer recovery may be possible by the use of special recovery kit for FTA card (Mas et al., 2007; McClure et al., 2009; Stangegaard et al., 2011) or use of alternative storage media (Allen-Hall & McNevin, 2013; Ivanova & Kuzmina, 2013; Lee et al., 2012). Whole genome amplification may be useful for generating suitable quantities of DNA from minute amounts (Pinard et al., 2006). At the same time, presence of foreign DNA in the mucus and its effect in downstream analyses should be investigated in the near future.

In conclusion, we demonstrated that mucus samples collected underwater can be effectively used for PCR based population genetic studies in manta rays. This newly described method may create new opportunities to study sensitive or threatened species in regions where tissue sampling had been discouraged or prevented previously. However, tissue sampling remains as the most preferred option for DNA sampling until more conclusive testing on yields and presence of foreign DNA are completed and for additional reasons that tissues are also useful for research applications such as fatty acid and stable isotope analyses (Couturier et al., 2013a; Couturier et al., 2013b).

Supplemental Information

Video S1 Mucus sampling using a small toothbrush held in the diver’s hand

Click here for additional data file.

We would like to thank the Proytecto Mantas Ecuador staff and volunteers for their assistance with collecting mucus samples in Ecuador. We would also like to thank Drs. Stephen Doblin, Kevin Smith and Kevin Dodson for their support of EAM. We thank Jenny Ovenden, Myrna Constantin, James Hereward and Ed Heist for arranging equipment access.

Additional Information and Declarations

Competing Interests

Author Contributions

Animal Ethics

Field Study Permissions

DNA Deposition

The authors declare there are no competing interests.

Tom Kashiwagi and Ana B. Christensen conceived and designed the experiments, performed the experiments, analyzed the data, contributed reagents/materials/analysis tools, wrote the paper, prepared figures and/or tables, reviewed drafts of the paper.

Elisabeth A. Maxwell conceived and designed the experiments, performed the experiments, analyzed the data, contributed reagents/materials/analysis tools, wrote the paper, prepared figures and/or tables, reviewed drafts of the paper, conducted field trips.

Andrea D. Marshall conceived and designed the experiments, performed the experiments, contributed reagents/materials/analysis tools, wrote the paper, prepared figures and/or tables, reviewed drafts of the paper, organised and conducted field trips.

The following information was supplied relating to ethical approvals (i.e., approving body and any reference numbers):

University of Queensland Animal Ethics Committee, approval number: SBMS/206/11/ARC.

The following information was supplied relating to field study approvals (i.e., approving body and any reference numbers):

Ecuadorian Ministry of the Environment, research permit number: 008RM-DPM-MA.

The following information was supplied regarding the deposition of DNA sequences:

GenBank accession numbers: KR73234–KR73237.

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
