# Peer review of "Evaluating manta ray mucus as an alternative DNA source for population genetics study: underwater-sampling, dry-storage and PCR success"

_PeerJ, doi:10.7717/peerj.1188_

## Round 0.1 · original submission · Major Revisions

The paper is interesting, however, the presented data are considered only preliminary. In order to improve the manuscript, the Authors must also take into account all the suggestions made in the 'validity sections' of the findings of the three reviewers.

·

Basic reporting

The English is good, the structure according to the template of the journal. The figure texts are adequate. However, important results are not shown as figures, tables or otherwise. This is stated by the authors to be a preliminary report and it is very much so. The results relevant to the hypotheses are not all clearly shown.

Experimental design

The scope is within the journak scope, meaningful research questions are stated (but not answered). Enough data on negative and positive controls, and results are not available, hence it is not possible to tell whether the investigation has been conducted rigorously and to a high technical standard.

Validity of the findings

This manuscripts sets out to describe 1) a method to do underwater sampling of mantra ray mucus, 2) dry-storage of the sample and 3) the success of PCR done on these samples.
The authors states that this is a manuscript showing preliminary results. The authors however fails to show even the preliminary results in a clear way. The paper shows that one can sample black mucus by the use of a toothbrush. However, when it comes to clearly show that dry storage and PCR are possible from this, a number of shortcomings are apparent.
*Figure 4 is showing the DNA absorbance in samples stored and extracted from FTA Elute cards. The results show that the absorbance spectra are more or less the same as what one get when using cards without mucus. It is unclear whether the 23.16 ng/µl DNA value stated in line 93 of the results is a real result. What would be the “DNA content” from unused (blank) cards? What wave length was used to calculate the DNA content? What was the blank used for the fluorometric measurements (Qubit)? What is the “DNA content” of blank cards using this (Qubit) method?
*The authors state that “PCR was successful for all five markers”, line 100-101. However, not a single PCR-result is shown. Would also elutes from blank cards give a positive PCR result? What was the negative control used?
*The authors states that “These PCR products were successfully sequenced and genotyped consisting with known types” line 101-102. However, not a single sequence is shown. In the M&M sections line 88 it is said that “the sequences are compared with known types”, however when showing no data this comparison cannot be done by readers of the manuscript.
So does the paper deliver the 3 points the title “promise”?
1) a method to do underwater sampling of mantra ray mucus. A black tooth brush and FTA elute card with mucus show that this is possible (figure 3). In discussion, line 116-118, it is said that tooth brush is the most effective, comparative results are however not shown
2) dry-storage of mucus sample possible (for further analysis). The results shown (fig 4) do not support this. Also other DNA content or PCR results to support this is not clearly shown.
3) the success of PCR done on these samples. The authors state to have PCR results, however such results are not shown.

Additional comments

To find a method to sample DNA from mucus for further dry storage and PCR analysis would be of value. However, to validate such a method clear data need to be provided. That is not the case for this manuscript.

·

Basic reporting

This study explores an alternative sampling methods for sharks and rays, by way of a swab-type approach with a toothbrush to sample mucus/DNA from animals, followed by the storage on Whatman FTA card. This pilot study demonstrates that DNA stored on the cards was reliable enough for the amplification of mtDNA an nDNA markers (ND5, RAG1 and microsatellites).

This manuscript adheres to the PeerJ guidelines

Experimental design

Although preliminary; I found this study very well written and highly relevant for sharks/rays conservation studies, allowing for a cost-efficient and rapid sampling of many animals, followed by successful analyses of standard genetic markers. Although FTA cards have already been used in many studies requiring rapid fixation of liquid or microscopic samples, this study seems to be the first to combine mucus sampling and FTA cards storage.

As stated by the authors, quite some improvements might be needed to leverage these DNA samples to the quality/quantity for population genomics work or even highly multiplexed microsatellite work (requiring higher yield/quality).

Validity of the findings

I have various remarks that should be addressed before this manuscript can be considered for publication in PeerJ:

• Although the Nanodrop profile is shown for a reference tissue sample, the overall quality of PCR, Sequencing reaction (how large were the fragments?) and microsatellite fragment profiles (including small-large fragment sizes) could have been shown. The authors only write that their analyses were successful without any kind of evaluable data to support this.

• It would have been more correct to jointly evaluate in parallel the mucus sampling + ethanol AND mucus + FTA cards to allow a comparison of the storage vs sample type. Result do not allow to discriminate between both variables to finetune the sampling/storage strategy.

• In line with my comment above, the Nanodrop results are quite worrying in fact, as there seem to be a whole lot of products (solvents, etc) from purification/storage and barely any DNA (confirmed by QUBIT). I fear that some residues will hamper future amplifications and a second cleanu-up might solve these problems. A test of mucus in ethanol would have pinpointed a mucus vs FTA elution cause for this pattern.

• In many other species, mucus has been known to contain a lot of PCR inhibitors. Although this didn’t play a role for current analyses, due to the lack of detailed results, future studies might underestimate this problem. A comparative analysis of inhibitor removal techniques would have been useful. This also holds for pigments in the mucus, potentially increasing the problem in more advanced analyses.

• When dealing with low concentration of quality samples, replicates should be included, as well as positive/negative controls. Although the latter ones seem to have been done, the former multiple amplifications are lacking, especially to evaluate genotype consistency.

Additional comments

As stated in the discussion, evaluating the amount and origin of foreign DNA in the mucus would be relevant, for instance by amplifying the 16s rDNA region to assess the mucus-associated microbiome interactions.

Reviewer 3 ·

Basic reporting

Largely okay, though there are some grammatical errors which need to be resolved. This will make the article less ambiguous (ln 28-29 for instance). Although the potential of this approach is clear my main concern is that there is no supporting evidence that the desired target sequences were amplified. In my opinion this data is paramount in proving the utility of the methodology and must be incorporated. If wordage is the limiting factor here the Introduction and Discussion could be trimmed.

Experimental design

The collection and quantification of DNA in the samples seems appropriate. However, I can see no evidence that amplifications were of the target species. The authors state that PCR of the targeted sequences from the target species were successful, but supply no supporting data for this. While I do not doubt the veracity of their findings they must be supported by BLAST scores or other indications of amplified sequence identity. Even successful amplification of focal species microsatellites is not evidence on its own. Also the full utility of the approach could be better assessed if evidence of microsatellite amplifications were shown. Is it really useful for this type of marker, or are there caveats?
The statement in the Discussion that better quality mucus samples were taken by circular brushing of the dorsal area is ambiguous. Are the authors suggesting the DNA from these samples is better quality, or just the volume of mucus and an indication there is something on the brush?

Validity of the findings

There is no doubt DNA has been included in the samples, that PCR amplified sequences of something, but there remains doubt that these were the desired sequences from the target species. The authors must have this data available and its inclusion would strengthen the ms considerably. Without it there is no confirmation that the approach is a useful contribution to population genetic studies.

Additional comments

An interesting and useful report, but requires some additional data to make it fully acceptable to the scientific community. The idea of transferring samples to FTA cards is an excellent idea, making the sampling process more amenable to routine sampling by non-specialists.

---

## Round 0.2 · accepted · Accept

The manuscript has been appropriately modified and improved to take into account all the suggestions proposed by the reviewers.